# Improved Enamel Acid Resistance by Highly Concentrated Acidulated Phosphate Sodium Monofluorophosphate Solution

**DOI:** 10.3390/ma15207298

**Published:** 2022-10-19

**Authors:** Ryouichi Satou, Atsushi Yamagishi, Atsushi Takayanagi, Miyu Iwasaki, Hideyuki Kamijo, Naoki Sugihara

**Affiliations:** 1Department of Epidemiology and Public Health, Tokyo Dental College, Tokyo 101-0061, Japan; 2Department of Social Security for Dentistry, Tokyo Dental College, Tokyo 101-0061, Japan

**Keywords:** fluoride, sodium monofluorophosphate, demineralization, preventive dentistry

## Abstract

Sodium monofluorophosphate (MFP) is a component of fluoride-containing dentifrices and is more biosafe than the conventional sodium fluoride (NaF). MFP can respond not only on the tooth surface layer but also deep into the enamel. We aim to confirm that high concentrations of acid phosphate MFP (AP-MFP, 9000 ppmF), used in professional care, could lead to a highly biosafe fluoride application method that acts through the deep enamel layers. Sample groups were respectively treated in vitro with NaF, acidulated phosphate fluoride (APF), MFP, and AP-MFP, and the samples were compared against an untreated group. Characterizations after fluoride application confirmed that MFP and AP-MFP treatments improved the acid resistance of enamel compared to that of conventional methods. Furthermore, the acid resistance of highly concentrated MFPs improved by using phosphoric acid. Although the acid resistance from the AP-MFP method is not as good as that using APF, AP-MFP can act both on the surface layer and deep into the enamel. Moreover, AP-MFP retains fluoride ions as much as APF does on the tooth surface. The proposed fluoride application method using AP-MFP introduces a dental treatment for acid resistance that is highly biosafe and penetrates deep layers of the enamel.

## 1. Introduction

The caries-preventing effect of fluoride has been clarified by many epidemiological and basic studies [1,2,3]. In fluoride application, sodium monofluorophosphate (Na_2_FPO_3_ or MFP) is primarily used in fluoride-containing dentifrices. MFP is a sodium salt represented by CAS No. 10163-15-2 [3] with a molecular weight of 143.95 g/mol. Dentifrices using the conventional sodium fluoride (NaF) and MFP have a statistically equivalent caries-preventing effect [4]. Meanwhile, MFP significantly increases the amount of fluoride incorporated into enamel in a concentration-dependent manner, and 1500 ppm of dentifrice suppresses demineralization and promotes remineralization [5]. 

Furthermore, MFP has an advantage in clinical applications because of the very low biotoxicity of fluorine in the complex ionic state. Shourie et al. reported that the biotoxicity of MFP is only approximately one-third that of NaF [6,7]. In addition, MFP is highly soluble. When the saturation concentration of NaF is approximately 4%, MFP can be liquefied at 10 times the concentration or more. In the case of composite MFP ions, three activated fluoride ions may be present based on the atomic weight of fluoride in the solution [7]. In the case of NaF, in an environment where a large number of calcium ions are present, free fluoride ions rapidly generate calcium fluoride, and the fluoride ion concentration significantly drops [8]. On the other hand, MFP maintains the structure of complex ions even in a solution containing calcium and can have at least 70 times more free fluoride ions as NaF [8]. Even in saliva and chemicals in which calcium ions are supersaturated, MFP can maintain a high fluoride ion concentration, which is one of the advantages of using MFP in dentifrices [9]. When MFP reacts with the tooth substance, its immediate effect after application is lower than that of NaF, and the acid resistance near the surface layer (0–50 μm) is also inferior [8]. However, owing to the difference of its action mechanism from that of NaF, MFP penetrates not only the surface layer of the dentin but also the deep enamel parts (50–300 μm). MFP may also form a thicker and more uniform acid-resistant layer than NaF [7,10,11].

Sodium fluoride has been shown to improve resistance to phosphate acidity, but there are no reports yet of MFP used in acidic conditions [1,2]. Despite its many advantages over NaF, the use of MFP is limited to dentifrices, and its concentration is limited to 1500 ppm [3]. We hypothesized that high concentrations of MFP, used in professional care for tooth surface application, could help develop a highly biosafe method that acts deep through the tooth substance. This study aims to develop a new fluoride application method by taking advantage of the high biosafety of MFP and their ability to respond to deep teeth layers. In addition, enamel acid resistance after the application of the proposed method was compared with that using conventional materials (NaF and APF).

## 2. Materials and Methods

### 2.1. Preparation of Enamel Samples

Forty bovine anterior mandibular teeth were used in this study. After removing the attached gingiva and cement, the root of each tooth was removed and only the crown enamel was used in the experiment. Enamel blocks (W 1 cm × D 1 cm× H 1 cm) were then prepared and mirror-polished with water-resistant abrasive paper (#1000, #2000, and #4000).

### 2.2. Fluoride Application and pH-Cycling Acid Challenge

We prepared two types of highly concentrated sodium monofluorophosphate (MFP) solutions for topical fluoride application. The first MFP solution was prepared using MFP powder (sodium fluorophosphate, CAS:10163-15-2, Na_2_PO_3_F, FUJIFILM Wako Pure Chemical Corp., Tokyo, Japan) and distilled water to obtain 0.476 M of MFP solution with a fluoride concentration of 9048 ppmF. The second is acidulated phosphate sodium monofluorophosphate (AP-MFP), which is a 0.476 M MFP solution prepared at pH 3.6 with phosphoric acid (CAS:7664-38-2, H_3_PO_4_, FUJIFILM Wako Pure Chemical Corp., Tokyo, Japan). The samples were divided into five groups: (1) sodium fluoride (NaF, 9048 ppmF, pH 7.0) for 4 min, (2) acidulated phosphate fluoride (APF, 9048 ppmF, pH 3.6) for 4 min, (3) sodium monofluorophosphate (MFP, 9048 ppmF, pH 7.0) for 4 min, (4) acidulated phosphate sodium monofluorophosphate (AP-MFP, 9048 ppmF, pH 3.6) for 4 min, and (5) untreated fluoride (Control, 0 ppmF, pH 7.0). Eight samples were prepared from each group (n = 8). To create an experimental and control surface on the same enamel surface, half of the mirror-polished enamel surface was coated with dental sticky wax. In this study, a pH-cycling test was conducted. After fluoride application for 4 min, all samples from the five groups were then immersed in a remineralization solution (0.02M HEPES based buffer solution, Ca: 3 mM, P: 1.8 mM, pH 7.3, DS: 10) for 1 h at 37 °C. After the remineralization treatment, samples were immersed in a demineralization solution (0.1 M lactic acid buffer solution, Ca: 3 mM, P: 1.8 mM, pH 4.5, DS: 10) for 6 h at 37 °C. Each cycle was repeated four times.

### 2.3. Step Height Profiles Using 3D Laser Microscopy

We used a 3D measurement laser microscope (LEXT OLS4000, Olympus, Tokyo, Japan) to measure the step height profile between the experimental and control surfaces after the acid challenge. This experiment measured the amount of substantial tooth defects due to acid challenge. The measurement area was set at 645 µm × 645 µm, and the boundary between the acid-demineralized experimental surface and wax-protected control surface was photographed. Three-dimensional (3D) measurements were performed at five sites for each sample, and the mean and standard deviation were determined.

### 2.4. Cross-Section and Surface Morphology Using Scanning Electron Microscopy

After the acid challenge, each group of samples was washed with xylene. The samples were then dehydrated using an ascending ethanol series. After the carbon had evaporated from the samples, the tooth surface was observed using a scanning electron microscope (SU6600, HITACHI, Tokyo, Japan) at an accelerating voltage of 15 kV. The samples were then embedded in polyester resin (Rigolac, Nisshin EM, Tokyo, Japan) to prepare polished sections, and the cross-sections were observed.

### 2.5. Mineral Loss and Demineralization Depth Using Contact Microradiography

The samples were embedded in polyester resin (Rigolac, Nisshin EM, Tokyo, Japan) to prepare 100 µm thick polished sections. Using a soft X-ray generator (CMR-3, Softex, Tokyo, Japan) equipped with a 20 µm thick Ni filter, imaging conditions were set to enable differentiation of the 20 steps of an aluminum step wedge, with a step height of 20 µm. Imaging was performed with a tube voltage of 15 kV, tube current of 3 mA, and radiation time of 12–15 min, and light microscopy was performed at 200× magnification. A glass plate (high-precision plate, HRP-SN-2; Konica Minolta, Tokyo, Japan) was used for the imaging. The plate was then placed in a developer (D-19, Kodak, Rochester, NY, USA) at 20 °C for 5 min. The plate was fixed for 5 min and washed with water for 10 min. The completed plates were converted to grayscale (8 bit, 256 tone) using an image analysis software (Image Pro Plus, version 6.2; Media Cybernetics Inc., Silver Spring, MD, USA) and an image analysis system (HC-2500/OL; OLYMPUS, Tokyo, Japan), and the concentration profile was acquired. The mineral loss value (ΔZ) and lesion depth (Ld) were measured, and the extent of demineralization was compared. Each of the five sites was measured in the range of 50 µm × 200 µm from the surface vicinity of the deep healthy enamel. For ΔZ, the mineral equivalent was calculated using the formula of Angmar et al. [12], using the density of the sample and the aluminum step wedge captured at the same time as the reference. The values were converted to a histogram with a mineral value of 0% and a healthy enamel section of 100% [13]. Lesion depth (Ld) was defined as the distance from the enamel surface to the location of the lesion where the mineral content was greater than 95% of that in sound enamel [13].

### 2.6. Surface Analysis by X-ray Photoelectron Spectroscopy

After fluoride application, each group of samples was dehydrated in an ascending ethanol series and polished to a thickness of 4 mm. Surface analysis was performed using an X-ray photoelectron spectroscopy (XPS) analyzer (XPS, AXIS-ULTRA, Kratos Analytical, Manchester, UK) under Al Kα, 15 kV, and 10 mA conditions. The spectrum of each element was corrected for the hydrocarbon-binding energy, where C1s = 285.0 eV. The absorbance was identified by wide scanning (binding energy of 1000 eV–0 eV) and each element was confirmed [14].

### 2.7. Statistical Analysis

The mean ± standard deviation (SD) of eight samples was determined to compare the five fluoride applications. The *p*-values were also calculated by one-way analysis of variance (ANOVA), and results were considered significant at *p* < 0.01. The Bonferroni test was used for post hoc comparisons when significance was determined using ANOVA (*p* < 0.01). Graphs were prepared and data were analyzed using the Origin software (ORIGIN 2022, Lightstone Corp, Tokyo, Japan).

## 3. Results

### 3.1. Step Height Profiles by 3D Laser Microscopy after the Acid Challenge

Figure 1 shows an image of the step profile obtained using the 3D laser microscope after the acid challenge. The left side of Figure 1A–E shows the reference surface (RS) protected by wax and not demineralized. The right side shows the demineralized experimental surface (ES). In the control group (no fluoride group), the ES was significantly demineralized, and a defect of 33.636 ± 2.962 μm was observed on the surface layer of the enamel (Figure 1a). In the NaF group, the difference in height between the RS and ES decreased to 1.895 ± 0.875 μm, and significant demineralization suppression was confirmed compared with the control group (no fluoride group) (*p* < 0.01) (Figure 1b). The height difference in the APF group (1.172 ± 0.594 μm) was smaller than that in the NaF group. However, no significant difference was observed between the NaF and APF groups (*p* > 0.01). The number of steps in the APF group was significantly smaller than that in the control, MFP, and AP-MFP groups, and the difference between the RS and ES groups was the smallest among the five groups (*p* < 0.01, Figure 1c). The step size of MFP was 5.633 ± 2.129 μm and that of AP-MFP was 4.206 ± 0.785 μm, and both groups suppressed demineralization more than the control group (Figure 1d,e). However, the two groups (MFP group and AP-MFP groups) using sodium monofluorophosphate as the fluoride component showed larger defects than the two groups (NaF group and APF groups) using sodium fluoride (Figure 1f).

### 3.2. Enamel Surface and Cross-Section SEM Observations after the Acid Challenge

Figure 2 shows the secondary electron image of the enamel surface after the acid challenge. The control group showed clear enamel rods and gaps (Figure 2a). A large number of fine spherical particles adhered to the surface and uniformly covered the surface of the NaF group (Figure 2b). Similar to the NaF group, spherical particles were uniformly generated in the APF group. However, the particle size was larger than that of the NaF particles, and the particle shape was not perfectly spherical (Figure 2c). In both MFP and AP-MFP groups, almost no particulate matter was found on the enamel surface and the diameter of the existing particles was small (Figure 2d,e). The surfaces of the enamel rods in both groups were rough, and some of the irregularities due to the enamel rod gaps disappeared. The images of the MFP and AP-MFP groups were very similar with no notable differences (Figure 2d,e).

Secondary electron SEM images of the cross-sectioned surfaces after the acid challenge are shown in Figure 3. In the control group, a decrease in signal intensity and collapse of the columnar structure were observed in the 0–20 μm range below the surface layer. In particular, serious surface demineralization was observed 5–10 μm below the surface layer (Figure 3a). In the NaF group, a uniform demineralization image within a range of 10 μm from the surface layer and subsurface demineralization due to the dissolution of scattered enamel prisms were observed at a depth of approximately 20 μm (Figure 3b). However, the NaF group maintained the trabecular structure and did not collapse compared with the control group (Figure 3a,b). The APF had a thin acid-resistant layer of 1–2 μm on the surface and a demineralized layer at a depth of 5–10 μm. No expansion of the enamel rod gap (inter-rod substance) was observed below 20 μm (Figure 3c). In the MFP group, as shown in Figure 3d, the trabecular structure collapsed from the surface layer to a depth of 10 μm, similar to that in the control group. However, a thick acid-resistant layer appears at 10–20 μm in the MFP group (Figure 3d). Below 20 μm, just below the acid-resistant layer, the signal strength of the enamel rod sheath increased because of dissolution of the central part of the enamel rod. In the MFP group, the demineralized and acid-resistant layers overlapped in multiple layers (Figure 3d). In the AP-MFP group, a surface layer of approximately 2 μm had irregularities due to demineralization, and a thin superficial demineralized layer was observed at a depth of 10 μm. However, no decrease in signal intensity was directly observed under the thick acid-resistant layer of 15–25 μm unlike that in the MFP group, and a signal was observed with the same intensity as that of the healthy part (Figure 3e).

### 3.3. Mineral Loss and Demineralization Depth by CMR Analysis

Figure 4 shows a CMR image of the boundary between the RS and ES on the enamel after the acid challenge. In the control group, a large defect due to demineralization was observed on the experimental surface on the right side of the image (Figure 4a). In the NaF group, the deficiency decreased, but multiple decalcification layers were formed in each cycle (Figure 4b). Moreover, demineralization was further reduced in the APF group compared to NaF, and the signal intensity was equivalent to that of healthy enamel, except for the demineralization of several micrometers on the surface layer (Figure 4c). The MFP group showed the second-largest defect after the control group, and multiple demineralized layers with the same number of cycles were observed as in the NaF group (Figure 4d). The AP-MFP group had defects in the surface layer, but the demineralized layer was smaller than that in the MFP group and was not multilayered. The signal intensity under demineralization was similar to that of the healthy enamel (Figure 4e).

Figure 5 shows the amount of mineral loss (ΔZ, vol% µm) and lesion depth (Ld, µm) in each group by CMR analysis (Figure 5a,b). In the control group, ΔZ was 11,430 ± 458 vol% μm, which was significantly higher than that in all other groups (*p* < 0.01, Figure 5a). In the NaF group, ΔZ decreased to 3971 ± 238 vol% μm, which was approximately 1/3 of that in the control group. The APF group was 1826 ± 659 vol% μm, which was approximately 1/6 of the control group. Meanwhile, the amount of ΔZ in the APF group was the lowest, which was significantly different from that in all groups (*p* < 0.01). The MFP group had 5353 ± 849 vol% μm and the AP-MFP group had 3490 ± 637 vol% μm, showing a significant difference between the two groups (*p* < 0.01). The lesion depth was the largest in the control group at 104.15 ± 7.84 μm, which was significantly different from that in the NaF, APF, and AP-MFP groups (*p* < 0.05, Figure 5b). The lesion depth (Ld) of the NaF group was 73.24 ± 10.45 μm, which was lower than that of the control group, and further decreased in the APF group to 57.12 ± 17.13 μm. Additionally, the Ld of the APF group was the smallest among all groups (*p* < 0.05). The MFP group Ld was 96.98 ± 7.72 μm, which was smaller than the control group, but no significant difference was observed between the two groups (*p* > 0.05). The Ld of the AP-MFP group was 76.02 ± 13.24 μm, which was almost the same as that of the NaF group.

### 3.4. Surface Analysis by XPS Analysis

Figure 6 shows the quantification of fluoride ions using XPS on the surface layer of enamel after various preventive measures. The binding energy of fluorine (1s orbital) was 685 eV, which is consistent with the peak values of CaF_2_ and fluorapatite. The peak intensity was 5909 cps in the control group, 6802 cps in the NaF group, and 16,107 cps in the APF group, which increased with the preventive treatment with sodium fluoride (Figure 6). In particular, the APF group exhibited the highest peak intensity (Figure 6). The MFP group showed the smallest value among other groups at 5344 cps, but it increased significantly to 14,753 cps in the AP-MFP group. The targets detected from the broad spectrum in the AP-MFP group were carbon, calcium, phosphorus, fluorine, nitrogen, and oxygen. The binding energy of calcium (2p_3/2_) in AP-MFP was 348 eV, which is consistent with the peaks of CaF_2_ and CaCl_2_. The binding energy of phosphorus is 134 eV, which is consistent with the peak of HAp. Furthermore, the binding energy of nitrogen was 400 eV, which is consistent with the amino group (-NH_2_) of proteins and collagen. Lastly, the binding energy of oxygen is 532 eV, which is consistent with the peaks of CaCO_3_ and HAp.

## 4. Discussion

### 4.1. Effectiveness of High-Concentration MFP-Based Tooth Surface Treatment Method

This study showed that treatment with MFP and AP-MFP improved enamel acid resistance similar to that in the NaF and APF groups. Based on the enamel loss and CMR results, the highly concentrated MFP (9000 ppm, pH 7.0) was the least effective in imparting acid resistance among the treatments in this study (Figure 1f). In addition, the amount of mineral loss and depth of demineralization of the MFP were inferior to those of the other three preventive treatments (Figure 5a,b). Cross-sectional SEM showed that the high-concentration MFP group had significant demineralization of the surface layer; however, a thick acid-resistant layer appeared 10–20 μm below the surface layer (Figure 3d). The appearance of an acid-resistant layer in the deep enamel and the demineralization image just below the acid-resistant layer are consistent with previous studies, suggesting penetration of the MFP into the deep enamel [9,10,11]. The 3D-measured laser microscopy height profiles and CMR mineral loss results showed stronger acid resistance enhancement in the AP-MFP group than in the MFP group (Figure 1f and Figure 5a). In the SEM cross-sectional images, the AP-MFP group showed an improvement in signal intensity from the surface layer to 10 μm compared to the MFP group, and the intensity deeper than 20 μm also improved to the same level as that of healthy enamel (Figure 3e). The present study revealed that high concentrations of MFP could qualitatively and quantitatively improve acid resistance by phosphate acidification (Figure 5a). This result suggests that the acid resistance of the fragile surface layer could be improved by changing the pH from 7.0 to 3.6, even if the fluoride ion concentration of the MFP was the same. In the case of sodium fluoride, intermediate dibasic calcium phosphate dihydrate (CaHPO_4_·2H_2_O, DCPD) has been reported to form in acidic environments, effectively incorporating F^−^ into hydroxyapatite [15]. Our results suggest that MFP also have a mechanism that promotes uptake in acidic environments, such as sodium fluoride. Quantification of the fluoride ions by XPS on the surface layer of the tooth showed that the amount of fluoride ions in the MFP was similar to that in the control group (Figure 6). On the other hand, the AP-MFP group increased the amount of fluoride ions to approximately 2.5 times that of the MFP group, which is close enough to the APF group (Figure 6). These results indicate that by making the MFP phosphate acidic, the MFP and enamel reacted, forming products containing fluoride ions, such as calcium phosphate, CaF_2_-like substances, and fluoroapatite. The formation of reactants is supported by the fact that the binding energy of fluorine (1s orbital) was 685 eV in the XPS analysis of AP-MFP, which is consistent with the peak values of CaF_2_ and fluoroapatite (Figure 6). Since no CaF_2_-like particles were observed on the MFP and AP-MFP in the surface SEM observations, the formation of fluorapatite is highly possible (Figure 2d,e). These results suggest that the enhancement by surface fluoride ions in AP-MFP can improve the weak acid resistance of the enamel surface layer, which is a disadvantage of MFP compared to NaF. Thus, the proposed AP-MFP is a new tooth surface treatment method that can approach both the surface and deep layers of the enamel.

### 4.2. Enamel Acid Resistance Mechanism of AP-MFP

In the case of NaF and APF, CaF_2_ and similar deposits (weakly bound fluoride) are generated and deposited on the enamel surface via double decomposition with hydroxyapatite [16]. This weakly binding fluoride acts as a reservoir for the gradual release of ppm-order F^−^, which improves enamel acid resistance [17,18]. In the surface SEM images, CaF_2_-like particles appeared in large quantities on the tooth surface layer of the NaF and APF groups (Figure 2b,c). In the cross-sectional SEM surface layer, the NaF and APF groups formed a thin acid-resistant layer on the surface layer (Figure 3b,c). In acidic environments, intermediate DCPD is formed, effectively incorporating F^−^ into hydroxyapatite [15]. This indicates that APF can improve acid resistance more efficiently than NaF and has an immediate effect. The effectiveness of APF has been proven by previous studies and clinical reports [1,2]. In this study, the APF group also showed the highest acid resistance among all experimental groups in terms of enamel loss, mineral loss, and demineralization lesion depth (Figure 1f and Figure 5a,b). The amount of fluoride ions in the tooth surface layer was also the highest in the APF group (Figure 6). However, in the cross-sectional SEM of the APF group, a severe demineralization layer existed immediately below the surface acid-resistant layer, and its effect did not reach the deep layer of the enamel (Figure 3b,c).

There are two possible mechanisms by which the application of MFP and AP-MFP on the tooth surface confers resistance to enamel acid. (1) MFP dissociates and exists as PO_3_F^2−^ but is degraded to F^−^ by phosphate hydrolysis enzymes in the oral cavity or by interaction with the surface of hydroxyapatite, and then acts in the same way as NaF [19,20]. In addition to the mechanism described above, (2) PO_3_F^2−^ replaces HPO_4_^2−^ in calcium-deficient hydroxyapatite [10,11]. Multiple substances are generated on the surface layer [8,9]. Tanizawa et al. performed electron spectroscopy for chemical analysis (ESCA) on the surface of hydroxyapatite after the application of highly concentrated MFP (10,000 ppm) and reported that CaF_2_ and CaPO_3_F were formed on the tooth surface layer [8]. Yamagishi et al. considered the formation of MFP-Ca salt, which is less water-soluble than CaF_2_, and the formation of MFP-ylated hydroxyapatite [Ca_10−x_(FPO_4_)_x_(PO_4_)_6−x_(OH)_2-x_XH_2_O], in which the HPO_4_ of hydroxyapatite is partially exchanged with MFP [9]. Because no plaque or saliva was used in the experiments of this study, there was no phosphate hydrolase. Therefore, the improvement in acid resistance in this study is mainly due to the action of PO_3_F^2−^, replacing HPO_4_^2−^ in hydroxyapatite, which is the second theory for acid resistance. Because a large number of CaF_2_-like particles were not observed from the surface SEM on the enamel surface of the MFP and AP-MFP (Figure 2d,e), the mechanism of theory (2) is further confirmed.

Although MFP is less acid-resistant on the surface layer than NaF or APF, it can penetrate deep into the tooth structure and demonstrate acid resistance [10,11]. These reports were consistent with the enamel deficiency in the AP-MFP group, SEM observations, and CMR findings (Figure 3 and Figure 5). In addition, AP-MFP has an excellent property of retaining fluoride ions equivalent to APF on the tooth surface (Figure 3, Figure 5, and Figure 6). This study proposes a mechanism by which MFP is made acidic by phosphoric acid to enhance the formation of CaPO_3_F and the substitution efficiency of HPO_4_^2−^ and PO_3_F^2−^ in hydroxyapatite for the formation of MFP-ylated hydroxyapatite.

### 4.3. Clinical Application and Challenges of AP-MFP

This study was conducted in vitro and demonstrated the behavior of saliva and plaque in the absence of MFP-degrading enzymes. There is no pellicle on the surface layer of the enamel, and AP-MFP action behavior may be different in the actual oral cavity where salivary proteins are present. In particular, the stability of AP-MFP in the oral cavity needs to be examined. The reaction of AP-MFP with tissues that are porous and contain a large percentage of organic matter, such as dentin, should also be investigated. In clinical practice, given the high biosafety of MFP compared to NaF, AP-MFP is suitable for children who have high fluoride bio-uptake and require deep penetration of fluoride ions into the tooth structure, such as in white spots. However, since MFP is inferior to sodium fluoride in terms of immediate effect and acid resistance of the enamel surface layer, a combination of APF and AP-MFP that can compensate for each other’s disadvantages is currently considered preferable. For instance, Mellberg et al. reported that a combination of NaF and MFP was more effective than a fluoride concentration of 300 ppmF from either agent alone [21]. Application methods that can act continuously on tooth surfaces for a long time, such as varnish, should be considered.

Although this experiment was done to prevent tooth caries, it might prevent erosion by strengthening the tooth structure. The application of fluoride is effective against erosion and is used in clinical dental practice. Rinsing with a fluoride and stannous chloride-containing mouthwash before an erosive attack reduced the softening of enamel [22]. Surface sealants are able to reduce the erosive dentine mineral loss [23]. Since erosion is influenced by multiple factors, including temperature and flow rate of the demineralizing solution, there is insufficient information based on this experiment alone [24]. However, an increase in the concentration of fluoride ions in deep layers of the enamel would certainly increase resistance to erosion.

In all preventive treatments in this study, the conventional protocol was followed for a 4-min application on the tooth surface using a cotton ball with liquid. However, MFP is less immediate than NaF, and its effectiveness can be further enhanced by extending the application time or making it into a gel. The appropriate application time and properties of AP-MFP must be clarified in future experiments. In addition, the hydrolysis of MFP occurs in acidic environments. In this study, AP-MFP was used within 30 min of preparation, but the stability of AP-MFP after adjustment and the possibility of long-term storage need to be examined. If unstable, the clinical procedure should be modified to include a pH adjustment by adding phosphoric acid immediately before application. In the future, it is desirable to examine the acid resistance of AP-MFP in dentin, its dynamics in the presence of saliva and plaque, and the appropriate recall period. Preparation of the basic data for clinical application is also recommended as the development of the proposed method progresses.

## 5. Conclusions

The proposed fluoride application method using AP-MFP introduces a dental treatment for acid resistance that is highly biosafe and penetrates deep layers of the enamel. Moreover, AP-MFP retains fluoride ions as much as APF does on the tooth surface. The AP-MFP demonstrated higher inhibition of demineralization than the APF in deep layers, both qualitatively and quantitatively. The proposed AP-MFP method is expected to become a new standard for providing professional care to prevent dental caries.

## Figures and Tables

**Figure 1 materials-15-07298-f001:**
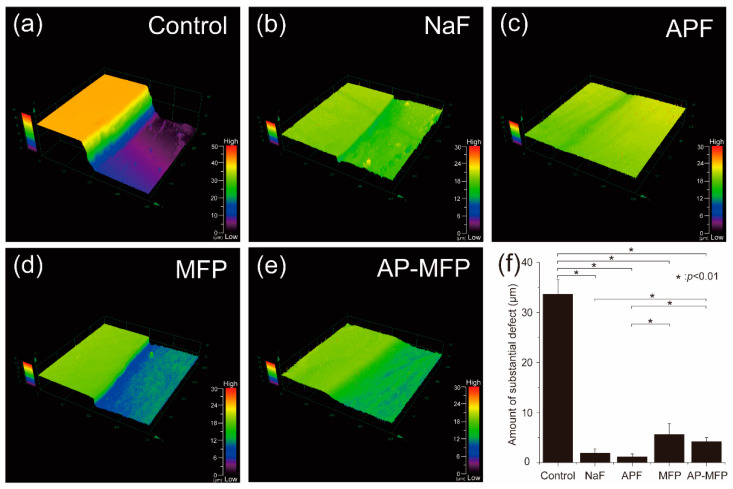
Height difference profiles measured using a 3D laser microscope. Boundary images of the reference and experimental surfaces of the samples from the (**a**) control (not fluoride treated), (**b**) NaF (9048 ppmF, pH 7.0), (**c**) APF (9048 ppmF, pH 3.6), (**d**) MFP (9048 ppmF, pH 7.0), and (**e**) AP-MFP (9048 ppmF, pH 3.6) groups after the acid challenge. The left side of Figure (**a**–**e**) shows the RS protected by wax and not demineralized. The right side shows the ES that has been demineralized. (**f**) Graphical representation of the substantial defects due to demineralization (*n* = 8, * *p* < 0.01).

**Figure 2 materials-15-07298-f002:**
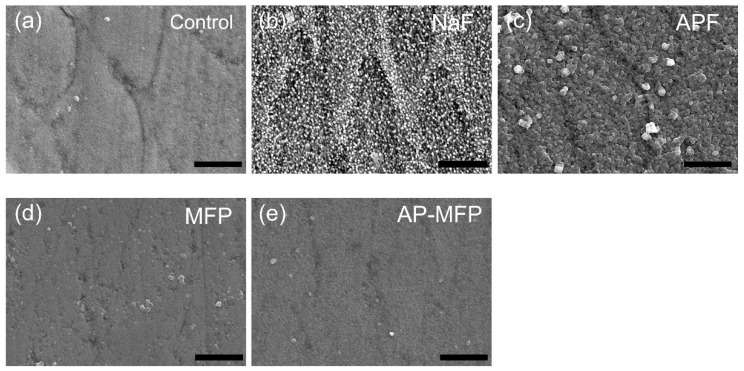
Surface SEM images of the samples from the (**a**) control, (**b**) NaF, (**c**) APF, (**d**) MFP, and (**e**) AP-MFP groups after the acid challenge. Scale bar is 2.5 μm. All images were recorded at 10,000-fold magnification.

**Figure 3 materials-15-07298-f003:**
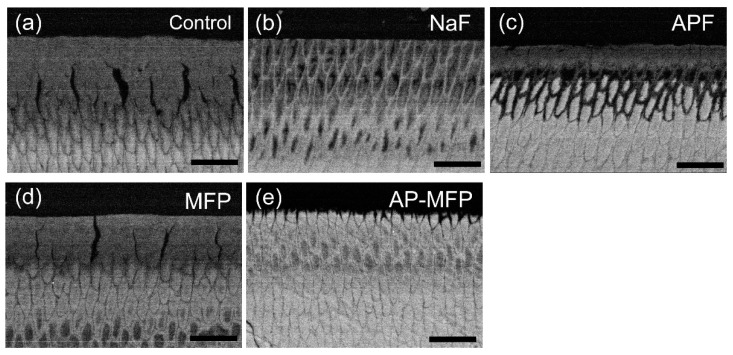
Cross-sectional SEM images of the samples from the (**a**) control, (**b**) NaF, (**c**) APF, (**d**) MFP, and (**e**) AP-MFP groups. Scale bar is 10 μm. All images were recorded at 2000-fold magnification.

**Figure 4 materials-15-07298-f004:**
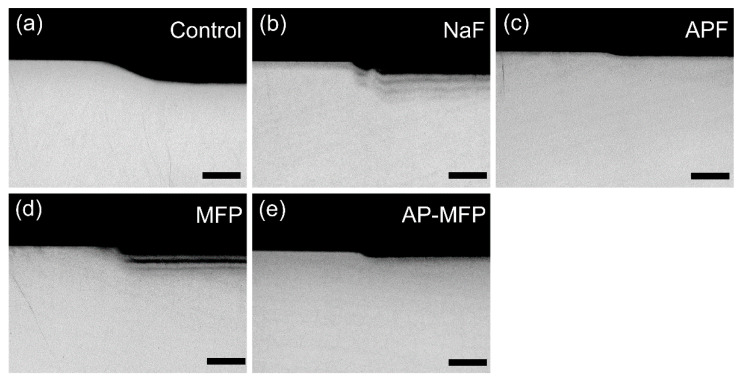
Boundary CMR cross-sectional images of the reference and experimental surfaces after the acid challenge: (**a**) control; (**b**) NaF; (**c**) APF; (**d**) MFP; and (**e**) AP-MFP groups. Scale bar is 100 μm. The left side of Figure (**a**–**e**) shows the RS protected by wax and not demineralized. The right side shows the ES that has been demineralized.

**Figure 5 materials-15-07298-f005:**
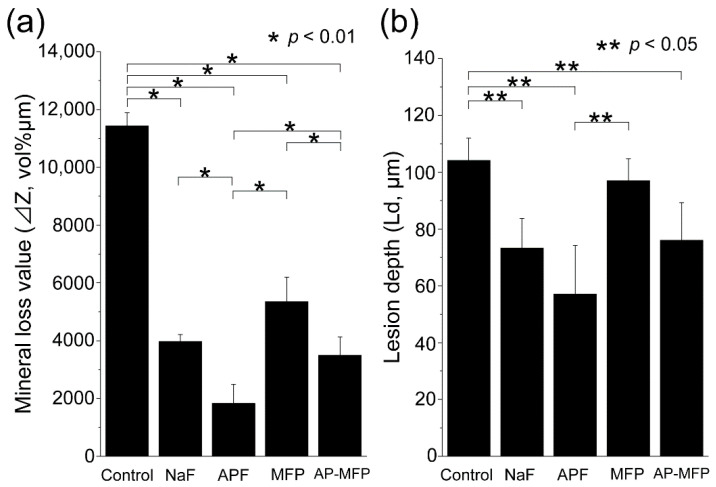
(**a**) Graphical representation of the mineral loss value (ΔZ, n = 8, * *p* < 0.01) after the acid challenge. For each sample, five sites were measured from the vicinity of the surface to the healthy enamel in a 50 × 300 μm region. All eight samples were measured and the mean ± SD is shown; (**b**) graphical representation of the lesion depth (Ld, n = 8, ** *p* < 0.05) after the acid challenge. From the surface prior to the demineralization experiment, the depth of demineralization was determined up to a site showing 95% healthy enamel. For each sample, five sites were measured. All eight samples were measured and the mean ± SD is shown.

**Figure 6 materials-15-07298-f006:**
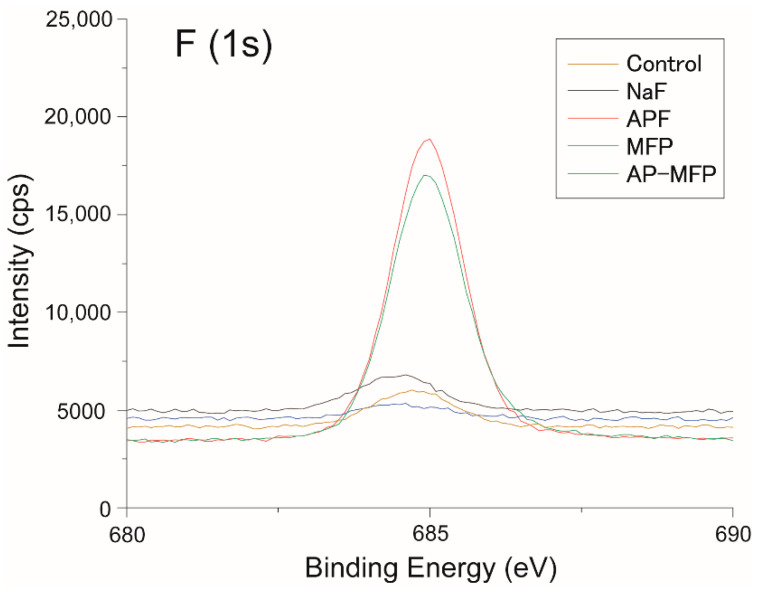
Surface analysis by XPS. The horizontal axis shows the binding energy (eV), and the longitudinal axis indicates the intensity (cps) of fluoride-ion 1S orbit. Yellow, black, red, blue, and green lines show the spectra for the control, NaF, APF, MFP, and AP-MFP groups, respectively. The spectrum of each element obtained was corrected by hydrocarbon-binding energy where C1s = 285.0 eV.

## Data Availability

All data are included in the manuscript.

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
