# Peer review of "Improved Enamel Acid Resistance by Highly Concentrated Acidulated Phosphate Sodium Monofluorophosphate Solution"

_materials, 2022, doi:10.3390/ma15207298_

Round 1
Reviewer 1 Report
The manuscript by Ryouichi Satou et al. is a well-prepared study, carried out by a complex of methods. All data are approved by experimental data. However, the Conclusion part summarizing the results is required. The manuscript can be accepted after minor revision.
Author Response
The manuscript by Ryouichi Satou et al. is a well-prepared study, carried out by a complex of methods. All data are approved by experimental data. However, the Conclusion part summarizing the results is required. The manuscript can be accepted after minor revision.
> We strongly appreciate the reviewer's comment. We are thankful for the time and energy you expended. In accordance with the reviewer's comment, we have added conclusion part as described below.
Page10-11, Line 399-405
- Conclusion
The proposed fluoride application method using AP-MFP introduces a dental treatment for acid resistance that is highly biosafe and penetrates deep layers of the enamel. Moreover, AP-MFP retains fluoride ions as much as APF does on the tooth surface. The AP-MFP demonstrated higher inhibition of demineralization than the APF in deep layers, both qualitatively and quantitatively. The proposed AP-MFP method is expected to become a new standard for providing professional care to prevent dental caries.
Reviewer 2 Report
Dear Author,
Very well written and relevant topic
One suggestion would be " how did you arrive at to sample size of 8 and is it compensate for the loss of sample during the study"
Author Response
Dear Author,
Very well written and relevant topic
One suggestion would be " how did you arrive at to sample size of 8 and is it compensate for the loss of sample during the study"
> Thank you very much for providing important comments. We are thankful for the time and energy you expended. We found that similar previous studies have been performed with n=5 to 15 (CMR and height difference profiles measured using a 3D laser microscope). We started with n=10 samples per group. However, handling 100μm thick samples for CMR was difficult and 1-2 samples in each group were broken. Therefore, the final number of samples was n=8.
Reviewer 3 Report
The article has been well written and does not require correction.
Author Response
> We strongly appreciate the reviewer's comment. We are thankful for the time and energy you expended.
Reviewer 4 Report
This in vitro study aimed to investigate the enamel acid resistance provided by various fluoride application methods including a highly concentrated acidulated phosphate sodium monofluorophosphate solution. The topic of the work is of interest, and the study has been well performed in large parts. Unfortunately, the authors only focused on the caries-preventing effect of fluoride. In order to improve the manuscript, the authors should at least discuss possible effects of the fluoride application methods on the acid resistance of dental hard tissues under more acidic (erosive) conditions. Possible erosion-preventive effects of fluorides should be discussed in a separate paragraph of the Discussion chapter of the manuscript by also taking into account other erosion-preventing measures (DOI: 10.3290/j.ohpd.b2259087, DOI: 10.3109/00016357.2012.757361, DOI: 10.1007/s00784-012-0731-3).
Author Response
>Thank you very much for providing important comments. We are thankful for the time and energy you expended. Our responses to the referees’ comments are as follow:
The reviewer's comment is correct. In accordance with the reviewer's comment, we have added following sentence in discussion section and references.
Page 10, Line 377-385
Although this experiment was done to prevent tooth caries, it might prevent erosion by strengthening the tooth structure. The application of fluoride is effective against erosion and is used in clinical dental practice. Rinsing with a fluoride and stannous chloride containing mouthwash before an erosive attack reduced the softening of enamel [22]. Surface sealants are able to reduce the erosive dentine mineral loss [23]. Since erosion is influenced by multiple factors, including temperature and flow rate of the demineralizing solution, there is insufficient information based on this experiment alone [24]. However, an increase in the concentration of fluoride ions deep layers of the enamel would certainly increase resistance to erosion.
References
- Philipp K.; Thanh P.N.; Blend H.; Thomas A.; Florian J.W. Enamel Softening Can Be Reduced by Rinsing with a Fluoride Mouthwash Before Dental Erosion but Not with a Calcium Solution. Oral Health Prev Dent. 2021, 7;19, 587-594; DOI: 10.3290/j.ohpd.b2259087
- Florian J.W.; Tobias T.T.; Thomas A. Durability of the anti-erosive effect of surfaces sealants under erosive abrasive conditions. Acta Odontol Scand. 2013, 71;5, 1188-94; DOI: 10.3109/00016357.2012.757361.
- Thomas A.; Klaus B.; Annette W.; Florian J.W. Impact of laminar flow velocity of different acids on enamel calcium loss. Clin Oral Investig. 2013, 17;2, 595-600; DOI: 10.1007/s00784-012-0731-3
Round 2
Reviewer 4 Report
In references 22-24, the first and last names are incorrectly reversed. Please correct.
Author Response
> The reviewer's comment is correct. We fixed the reference 22-24.
References
- Körner P.; Nguyen T.P.; Hamza B.; Attin T.; Wegehaupt F.J. Enamel Softening Can Be Reduced by Rinsing with a Fluoride Mouthwash Before Dental Erosion but Not with a Calcium Solution. Oral Health Prev Dent. 2021, 7;19, 587-594; DOI: 10.3290/j.ohpd.b2259087
- Wegehaupt F.J.; Tauböck T.T.; Attin T. Durability of the anti-erosive effect of surfaces sealants under erosive abrasive conditions. Acta Odontol Scand. 2013, 71;5, 1188-94; DOI: 10.3109/00016357.2012.757361.
- Attin T.; Becker K.; Wiegand A.; Tauböck T.T.; Wegehaupt F.J. Impact of laminar flow velocity of different acids on enamel calcium loss. Clin Oral Investig. 2013, 17;2, 595-600; DOI: 10.1007/s00784-012-0731-3